# Modulating the strong metal-support interaction of single-atom catalysts via vicinal structure decoration

Jingyi Yang[1,7], Yike Huang 🄳 [1,2,7], Haifeng Qi 🄳 [1], Chaobin Zeng[3], Qike Jiang[4], Yitao Cui[5], Yang Su[1], Xiaorui Du[1], Xiaoli Pan[1], Xiaoyan Liu 🄳 [1], Weizhen Li 🄳 [1✉], Botao Qiao 🄳 [1,4✉], Aiqin Wang 🄳 [1] & Tao Zhang 🄳 [1,6]

Metal-support interaction predominately determines the electronic structure of metal atoms in single-atom catalysts (SACs), largely affecting their catalytic performance. However, directly tuning the metal-support interaction in oxide supported SACs remains challenging. Here, we report a new strategy to subtly regulate the strong covalent metal-support interaction (CMSI) of Pt/CoFe$_2$O$_4$ SACs by a simple water soaking treatment. Detailed studies reveal that the CMSI is weakened by the bonding of H$^+$, generated from water dissociation, onto the interface of Pt-O-Fe, resulting in reduced charge transfer from metal to support and leading to an increase of C-H bond activation in CH$_4$ combustion by more than 50 folds. This strategy is general and can be extended to other CMSI-existed metal-supported catalysts, providing a powerful tool to modulating the catalytic performance of SACs.

[1] CAS Key Laboratory of Science and Technology on Applied Catalysis, Dalian Institute of Chemical Physics, Chinese Academy of Sciences, Dalian 116023, China. [2] University of Chinese Academy of Sciences, Beijing 100049, China. [3] Hitachi High Technologies (Shanghai) Co., Ltd, Shanghai 201203, China. [4] Dalian National Laboratory for Clean Energy, Chinese Academy of Sciences, Dalian 116023, China. [5] Synchrotron Radiation Laboratory, Laser and Synchrotron Research Center (LASOR), The Institute for Solid State Physics, The University of Tokyo, 1-490-2 Kouto, Shingu-cho Tatsuno, Hyogo 679-5165, Japan. [6] State Key Laboratory of Catalysis, Dalian Institute of Chemical Physics, Chinese Academy of Sciences, Dalian 116023, China. [7] These authors contributed equally: Jingyi Yang, Yike Huang. ✉email: weizhenli@dicp.ac.cn; bqiao@dicp.ac.cn

Metal-support interaction (MSI) plays a critical role in determining the catalytic performance (activity/selectivity/stability) of supported metal catalysts[1–3]. It is particularly true for single-atom catalysts (SACs) which are emerging as a new type of heterogeneous catalysts and have attracted extensive attention in the past decade due to their unique activity and/or selectivity[4–6]. In traditional nanocatalysts, the size and morphology of nanoparticles usually vary with different MSI[7–9]. The electronic state of the low-coordinated atoms which are located at the metal surface, terrance and kink sites far away from the metal-support interface and often serve as key active sites to adsorb reactant molecular and stabilize intermediate are, however, generally less influenced by the MSI. By contrast, in SACs all metal atoms are singly dispersed on, and directly interact with support, which not only maximizes the metal-support interfacial sites[10,11], but also renders the electronic properties and subsequently the catalytic performance of single atoms largely, or even chiefly, depending on the MSI[12–14]. Given that the metal atoms predominately interact with its surrounding support in static state, their electronic state, as well as catalytic performance, is therefore greatly dependent on their local coordination structure[15,16]. This has been well demonstrated by the established N-doped carbon supported SACs where the activity and/or selectivity can change significantly by varying the number of N-coordination[17–20].

However, the regulation of MSI on oxide-supported SACs by directly tuning the coordination structure is rather challenging on account of the robust crystalline structure of oxide support. Varying the metal oxide supports can certainly change the MSI and the electronic properties of metal atoms[21,22]. However, in this case the catalysts themself have changed thus the catalytic performance changes accordingly because the influence of properties of oxide support cannot be ignored at most of time, making the regulation out of control. Besides, tailoring the number of metal-oxygen bonds can be achieved by $H_2$-reduction treatments or other strategy on reducible oxides[23,24], whereas often at the cost of destabilizing the catalyst, arousing aggregation of single atoms. Therefore, strategies to regulate the MSI as well as the catalytic performance of oxide-supported SACs without the compromise of catalyst stability are highly desired.

In this work, we report a novel strategy to subtly modulate the strong covalent metal-support interaction (CMSI) of Pt SACs by tuning the vicinal enviroment of Pt single atoms via a simple water treatment without changing the dispersion and stability of Pt atoms. Detailed characterization demostrated that the water-soaking treatment can weaken the CMSI between Pt and $CoFe_2O_4$ support and reduce the chemical state of Pt, thus promote the activation of C-H bond towards methane combusion by more than 50 folds. DFT calcuation combined with control exmperiment revealed that the coordinaion of $H^+$, generated from water dissociation on the catalyst surface, at the Pt-O-Fe interface elongated the Pt-O bond and weakened the bonding strength accordingly. This strategy is general and can be extended to other catalysts systerm having CMSI, providing a simple yet effective avenue to modulate the catalytic performance of SACs.

## Results and discussion

**Structure characterization of $Pt_1$/CF SAC.** $CoFe_2O_4$ supported Pt SAC was synthesized by incipient wet impregnation of $H_2PtCl_6·6H_2O$ onto $CoFe_2O_4$ with a nominal Pt loading of 1 wt% followed by drying at 60 °C and calcination at 500 °C for 5 h, respectively, denoted as $Pt_1$/CF. Water treatment of $Pt_1$/CF SAC was performed by dispering 0.1 g catalysts into 100 ml ultrapure water for 2 h, then filtered and dried at 80 °C for 12 h. The as-pretreated sample is labeled as $Pt_1$/CF-W. The X-ray diffraction (XRD) patterns (Fig. S1) excluded the change of $CoFe_2O_4$ spinel phase after loading of Pt as well as after water treatment. It also suggests a high dispersion of Pt species due to the absence of any Pt diffraction peaks. High-angle annular dark-field scanning transmission electron microscopy (HAADF-STEM) images of $Pt_1$/CF and $Pt_1$/CF-W shown in Fig. 1 and S2 display that the Pt species existed exclusively as single atoms, suggesting that water treatment has no influence on the dispersion of Pt. Here the strong CMSI between Pt and Fe is critical to the formation of Pt single atoms, which has been elucidated in our previous studies[25].

CO adsorption diffuse reflectance infrared Fourier transform (CO-DRIFT) spectroscopy was applied to further confirm the structure of Pt species on both samples due to the sensitivity and accuracy of this technique in determining the electronic properties and isolation nature of Pt[26]. The absence of CO bridged adsorption in the region of 1870-1840 $cm^{-1}$ on $Pt_1$/CF (Fig. S3), together with the fact that CO linear adsorption peaks on Pt (2110 $cm^{-1}$ and 2088 $cm^{-1}$) are independent of CO coverage (Fig. 2a), unambiguously demonstrated the isolated single-atom dispersion of Pt on support[27,28]. These two peaks can be assigned to CO adsorbed on Pt single atoms with different valence states, and the one located at 2088 $cm^{-1}$ increased gradually with adsorption time evaluation, suggesting a reduction of Pt occurred during CO adsorption process[29]. The low intensity of CO adsorption is a typical character of CMSI which will result in total disappearance of CO adsorption upon calcination at higher temperatures such as 800 °C[25]. On $Pt_1$/CF-W sample, same adsorption character was observed except the increased adsorption intensity and decreased CO stretching frequency (Fig. 2a). It confirmed the isolation nature of Pt on the water-treated samples thus excluded the potential size effect raised by water treatment, in good agreement with the HAADF-STEM characterization result. It should be noted that the quantity of CO adsorption was significantly increased from 2 µmol/g to 19 µmol/g based on the CO chemisorption experiment (Table S1) and the frequency of CO stretching was decreased to ~2105 and 2084 $cm^{-1}$, respectively. Despite that, the CO adsorption quantity of $Pt_1$/CF-W was still much lower than the theoretical value (~50 µmol/g), illustrating that it is irrelevant to the dispersion of Pt. Ion scattering spectroscopy (ISS) evidenced that no significant change of Pt:Fe atomic ratio in the outermost atom layer after water

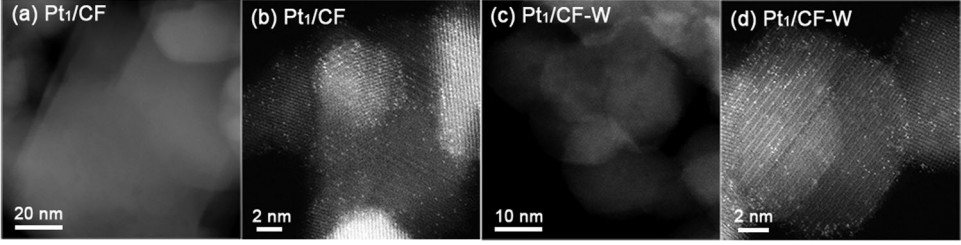

**Fig. 1 HAADF-STEM images. a**, **b** $Pt_1$/CF, **c**, **d** $Pt_1$/CF-W.

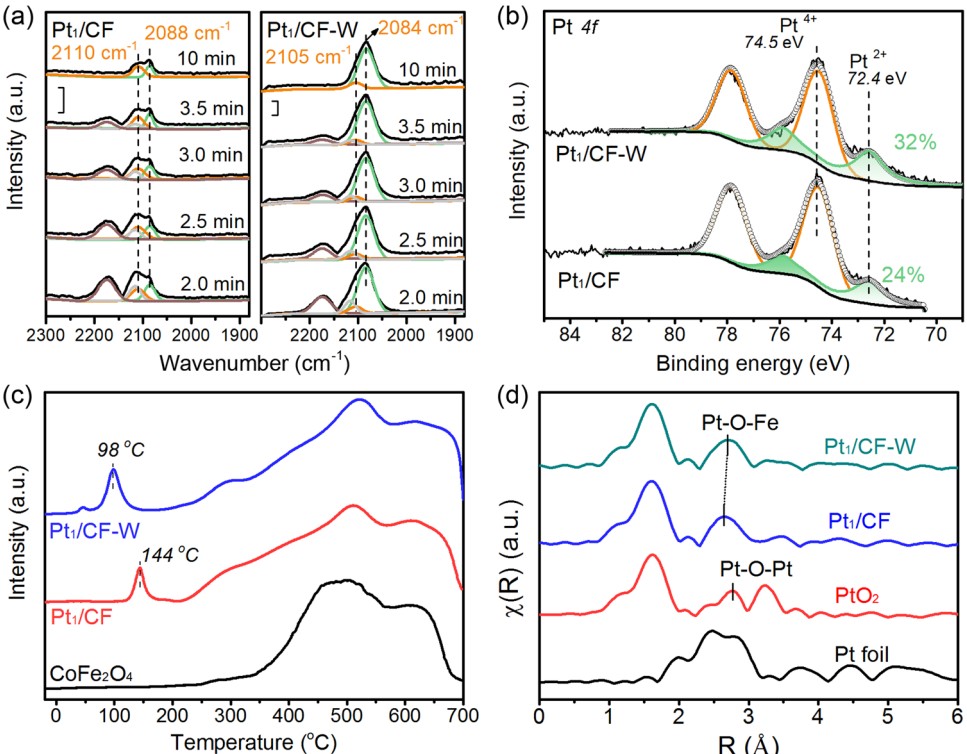

**Fig. 2 The MSI of Pt₁/CF and Pt₁/CF-W. a** Time-resolved CO-DRIFT spectra of Pt₁/CF and Pt₁/CF-W when purging the CO with He. **b** Pt *4 f* XPS spectra. **c** H₂-TPR profile. **d** The Fourier-transformed k²-weighted EXAFS spectra in R-space.

treatment, basically excluding the possibility that the change of CO adsorption capacity might originate from the different density of Pt single atoms on the surface (Fig. S4). Based on the above discussion, the change of intensity and frequency of CO adsorption on Pt₁/CF-W must have been a result of a weakened CMSI of Pt-CoFe₂O₄ along with a lower oxidation state of Pt[30].

X-ray photoelectron spectroscopy (XPS) of Pt *4f* spectra further confirmed the chemical state change of Pt single atoms on Pt₁/CF where Pt mainly existed as $Pt^{4+}$ (~76%) with a small quantity of $Pt^{2+}$ (24%) due to the strong interaction with CoFe₂O₄ support[31]. After water treatment, an increase of $Pt^{2+}$ ratio (32%) was observed on Pt₁/CF-W (Fig. 2b), indicating a relatively higher electron density of Pt single atoms resulted from the reduced charge transfer from Pt to CoFe₂O₄ support[32,33]. The weakened CMSI was further verified by H₂ temperature-programmed reduction (H₂-TPR) where the reduction of Pt species shifted to a distinctly lower temperature (144 to 98 °C) after water treatment (Fig. 2c)[34]. The quantitative analysis of H₂-TPR of the first peak showed that the consumption of H₂ in both cases are much higher than the theoretical one assuming $Pt^{4+}$ is reduced to $Pt^{0}$, illustrating that a large amount of support was reduced simultaneously due to the H₂ spillover. The slightly higher H₂ consumption of Pt₁/CF-W might be related to the increased ability of H₂ activation on the Pt single atoms with decreased valence state as suggested by the XPS (Table S2).

In order to understand the water-treatment effect on the MSI of Pt₁/CF in atomic scale, we evaluated the coordination environment of Pt single atoms by X-ray absorption spectroscopy (XAS). The X-ray absorption near-edge structure (XANES) of Pt₁/CF-W showed a slightly lower white line intensity in comparison with that of Pt₁/CF, suggesting a lower electronic state of Pt single atoms (Fig. S5), corroborating the CO-DRITF and XPS results[35]. The Fourier-transform extended X-ray absorption fine-structure (EXAFS) analysis revealed that Pt₁/CF and Pt₁/CF-W have similar Pt-O coordination numbers in the first shell (Fig. 2d,

Table S3). The absence of any prominent scattering peaks at the positions of either Pt-Pt (2.76 Å) or Pt-O-Pt (3.12 Å) on both catalysts in the detailed wavelet transform EXAFS (WT-EXAFS) analysis confirmed the sole presence of Pt single atoms (Fig. S6)[36], which is consistent with the HAADF-STEM and CO-DRIFT spectra results. However, Pt₁/CF and Pt₁/CF-W exhibited slightly different features at their second shell where Pt₁/CF-W showed a longer Pt-O-Fe distance than that on Pt₁/CF, suggesting that the vicinal coordination environment of Pt single atoms were modified during the water treatment. Because of the XAS is a characterization technique giving overall information of the catalyst entity, which means it is not sensitive to the surface and it cannot be denied that some of the Pt single atoms are anchored on the subsurface of CoFe₂O₄[25], the model of Pt₁/CF after water treatment was further studied in the following DFT simulations.

**Catalytic performance of Pt₁/CF SACs in methane combustion.** It is acknowledged that MSI plays a vital role in determining the catalyst performance in many reactions. Especially, it was proposed that a suitable MSI is critical in methane combustion[8,37], a reaction that is important for applications in clean energy conversion and environmental remediation[38,39]. In our recent study, Pt single atoms having CMSI with iron oxide support exhibited much improved methane combustion activity compared with their NP counterpart, whereas the total conversion temperature is still limited with a $T_{100}$ higher than 600 °C[25]; so did a recently reported Pt₁/Mn₂O₃[40]. The benefit of high oxygen mobility on such reducible oxides cannot be exploited under the oxygen-rich conditions[41,42] and the activities are suppressed on the highly oxidized Pt species due to the too strong MSI[43,44].

In this study, by simple water-soaking treatment we have modulated the CMSI and the chemical state of Pt which might be effective to modulate the activity. To examine our assumption, the catalytic activity of Pt₁/CF and Pt₁/CF-W towards methane combustion were tested in lean-burn condition under 0.5 vol%

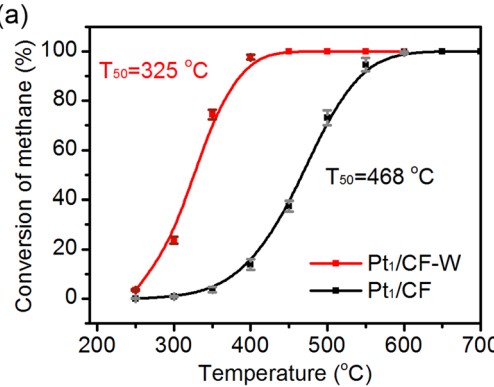
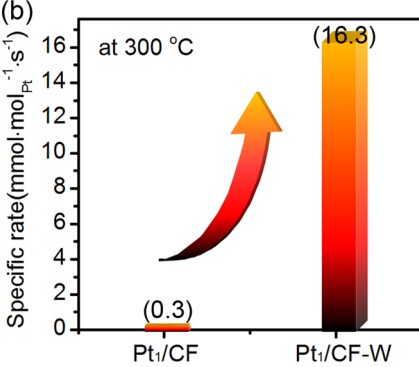

**Fig. 3 Catalytic activity of methane combustion. a** Light-off curves of $CH_4$ lean-burn combustion over $Pt_1$/CF and $Pt_1$/CF-W. Reaction conditions: 0.5 vol% $CH_4$, 20 vol% $O_2$, $N_2$; Gas hourly space velocity (GHSV) = 20,000 ml/($g_{cat}$·h). The error bars represent the standard deviation based on four repeated light-off experiments. **b** Specific rate of $CH_4$ conversion on $Pt_1$/CF and $Pt_1$/CF-W at 300 °C.

$CH_4$, 20 vol% $O_2$ and $N_2$. As shown in Fig. 3a, the as-prepared $Pt_1$/CF catalyst showed a $T_{50}$ (temperature for 50% conversion of methane) above 465 °C and a $T_{100}$ (temperature for total conversion of methane) as high as 650 °C, comparable to other Pt-based catalysts reported in literatures including Pt SACs[40,45,46]. However, after water treatment an astonishing activity increase was observed with dramatic drop of $T_{50}$ and $T_{100}$ for about 140 °C and 200 °C on $Pt_1$/CF-W catalyst, respectively. The reaction rate measured at 300 °C shows the activity increased by more than 50-fold after water treatment (Fig. 3b), and about 6–20 times higher than that of the most active Pt-based catalysts reported so far[27,47,48], and only inferior to the very recently reported $Pt_1$/$CeO_2$-$Al_2O_3$ catalyst[24] (Table S4). The activity is even comparable to those of the best Pd-based catalysts[49–53], as evidenced by the similar reaction curve on a homemade 1 wt% $Pd/Al_2O_3$ catalyst (Figs. S7, 8). The activation energy was 92 kJ/mol for $Pt_1$/CF catalysts while it decreased to 77 kJ/mol for $Pt_1$/CF-W (Fig. S9). As a control, $CoFe_2O_4$ support showed limited conversion at temperature below 400 °C and negligible activity change after identical water treatment process (Fig. S10), suggesting the contribution of water treatment must have originated from the change of Pt and/or MSI rather than $CoFe_2O_4$ support itself. HAADF-STEM images and CO-DRIFT spectra of the post-reaction $Pt_1$/CF-W confirmed that no structure change, neither dispersion nor chemical state, of Pt occurred during reaction (Fig. S11, 12).

**Verifying the origination of catalytic performance promotion.** Despite the fact that the promotional effect of water on the reaction mechanism has been reported in a few reactions such as CO oxidation through generating reactive hydroxyl, or forming $H_2O$-derived intermediate[54–56], for $CH_4$ combustion it is generally accepted that $H_2O$ has negative effect on the activity[57]. Therefore, it is less possible that the improved activity comes from the adsorbed $H_2O$ or generated $OH^-$ group directly participating into the reaction. In fact, if the adsorbed water molecular or OH- group generated from water dissociation participates in certain reaction step, they will be consumed quickly, resulting in a rapid catalyst deactivation[58]. However, the relatively good stability together with the fact that water in the feed gas actually inhibits the activity can exclude this possibility (Figs. S13, 14). To further verify this, we have treated the $Pt_1$/CF by isotope labeled $H_2^{18}O$ with an identical procedure which exhibited same activity promotion scenerio (Fig. S15). It reveals that only trace amount of $C^{16}O^{18}O$ was detected in the total $CO_2$ production (Fig. S16a), excluding the major role of $CH_4$ reacting with adsorbed $H_2^{18}O$ or $^{18}OH$- group. In addition, $CH_4$-TPR experiment on the $H_2^{18}O$ treated $Pt_1$/CF

sample also failed to detect any $^{18}O$-labled products (Fig. S16b), verifying that the O in the product comes from the support rather than the adsorbed $H_2O$ or generated hydroxyl group. Therefore water or the generated $OH^-$ doesn't contribute to the distinct activity increase by directly participating in the reaction.

One may argue that the activity increase might originate from the removal of $Cl^-$ by water treatment. We have therefore prepared a control catalyst by using $Pt(acac)_2$ as precursor (denoted as Pt(Ac)/CF-500), and all other procedures were kept unchanged. Activity promotion was still observed after water treatment (Fig. S17). In addition, the water-treatment effect has been found to be insensitive to the way of water extraction. We have volatilized all the treating water instead of filtration to avoid the possible change of counter ion, and the same promoting effect was observed (Fig. S18a), further excluding the impact of $Cl^-$. To avoid other soluble impurities in the commercial $CoFe_2O_4$, the oxide was washed with a large amount of water and calcinated at 500 °C before being used as a support. The fact that the scenario is similar as before suggests that the impurities such as alkali ions do not contribute to the activity increase (Fig. S18b).

After ruling out the above factors, size effect is a simple and easy come-to-mind explanation for the water-treatment effect since some literatures suggested that the aggregation of Pt single atoms gave rise to better activity[59]. However, by combination of HAADF-STEM, CO-DRIFTs and EXAFS techniques, we have excluded the potential influence of water treatment on the size of Pt. We also provided the activity data of $Pt_1$/CF with low loading (0.5 wt%) to examine the water treatment effect on the single atoms (Figs. S19, 20). The measured specific rate of 0.5 wt% $Pt_1$/CF and $Pt_1$/CF-W were similar to that of 1.0 wt% $Pt_1$/CF and $Pt_1$/CF-W, well illustrating that the activity is originated from the single atoms.

In addition, provided that Pt clusters contribute to higher reactivity, the activity of subnanometre Pt/CF catalysts prepared by identical procedure with higher Pt loading (up to 2 wt%) should show better activity than $Pt_1$/CF SACs, which is however not the case (Figs. S21, 22). Therefore, the contribution of potential aggregation of Pt single-atoms arising from water treatment can be completely excluded. On the other hand, the EXAFS, XPS and CO-DRIFTs characterizations suggested that the coordination microenvironment and electronic properties of Pt single atoms have changed slightly upon water treatment, which induced change of the CMSI on $Pt_1$/CF catalysts and may play a critical role in determining their catalytic performances.

**Importance of water on the vicinal environment of Pt single atoms.** To unravel the water treatment on the structure of Pt/CF, the plausible changes of surface O/Fe/Co properties were

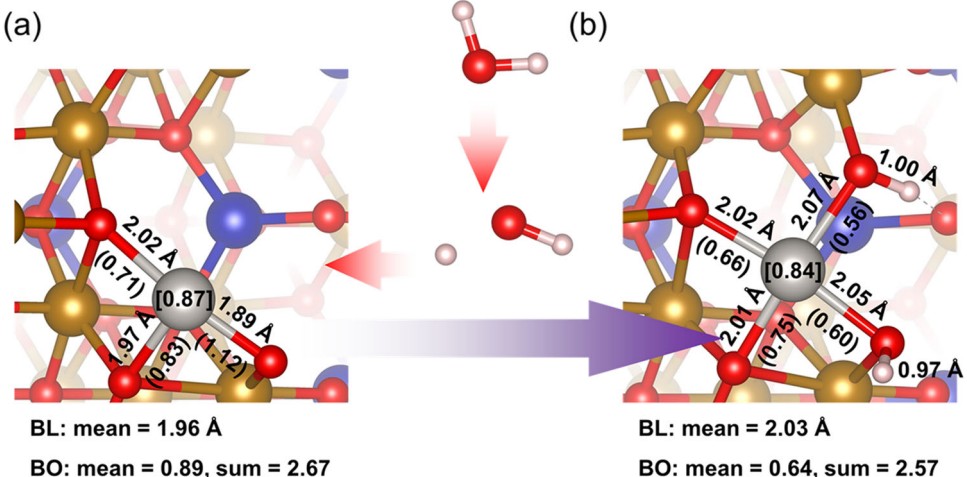

**Fig. 4 The structure decoration around Pt single atoms. a** Structure before water treatment, Pt/CF(110). **b** Structure after water treatment, Pt/CF(110)-W. BL: Pt-O or O-H bond length, BO (numbers in parentheses): Pt-O bond order. BL and BO are quoted beside the bond they belong to, numbers in square brackets are Bader charges of Pt atoms. Colors of atoms: H: white, O: red, Fe: brown, Co: blue, Pt: silver.

examined by XPS analysis. The O1s spectra show that no detectable changes of adsorbed water (533 eV) and lattice oxygen (529.7 eV) were observed on Pt/CF and Pt/CF-W (Fig. S23c). It is obvious that the quantity of $OH^-$ group (531.3 eV) on Pt/CF-W is more than that on Pt/CF, which must have come from the dissociation of water. The shift of both Fe 2p and Co 2p spectra to higher energy together with a detailed fitting analysis according to Biesinger et al.[60] further confirmed the hydroxylation of $FeO_x$ or $CoO_x$ by water (Fig. S23a, b, Table S5).

Since the results above indicated small structure de-decoration/decoration around Pt single-atom was made during the water treatment, atomic simulation was then employed to identify the possible structures of $Pt_1$/CF before and after water treatment. Here, Pt/CF(110) is modeled to represent structure of $Pt_1$/CF, based on the high-resolution STEM observation which revealed that Pt single atoms are mainly located on the $CoFe_2O_4$(110) facet (Fig. S24). For modeling $Pt_1$/CF-W, two kinds of structural decoration were considered: the water adsorption in molecular state ($H_2O$) or dissociation to hydroxyl ($OH^-$) and hydrogen ($H^+$) species on the catalyst surface[61]. After comparing energies between different configurations, it turned out that structure shown in Fig. 4b had the lowest energy and is denoted as Pt/CF(110)-W in following discussion.

Population (Bader charges)[62–65] and Mayer bond order analysis[66] were performed. As shown in Table S7, charge of Pt single atom slightly decreased from 0.87 to 0.84 $e$, which was consistent with our observation in XPS and XANES. Pt-O bond elongated from 1.96 to 2.03 Å on average, Mayer bond order decreased from 0.89 to 0.64 on average, indicating a weakened CMSI, which also reflected on a decreased reduction temperature of Pt in $H_2$-TPR experiment. In Pt/CF(110)-W, there were two O atoms coordinating with three atoms (H, Pt and Fe), and the bonds formed between O and Pt had the longest length and smallest bond order among the four Pt-O bonds. In these cases, coordination bonds where electrons transferred from O atom lone-pair (LP) electrons orbital to empty orbitals of surrounding atoms formed (One O-H bond had length 1.00 Å, similar to that of $H_3O^+$, 0.99 Å, implying a mixing between two electron transfer modes $LP(O) \rightarrow X$ and $X \rightarrow 2p$ (O), where X represents Pt, Fe and H. The other O-H bond had length 0.98 Å, similar to that of $H_2O$, 0.97 Å, implying a similar mixing but mostly happened merely in bonds Pt-O and Fe-O). Therefore the critial role of water on the moduation of vicinal environment of Pt single

atoms probably realized through the generated $H^+$ from water dissociation. To furhter verify this, a control experiment was performed where we observed that the enhanced activity is closely related to the existence of $H^+$ because less activity increase was observed when the ultrapure water was substituted by NaOH solutions to lowering the concentration of $H^+$ during the treatment process (Fig. S25). Based on the above discussion, the decoration of $H^+$ at the Pt-O-Fe interface during the water treatment is well depicted, which results in a reduced electronic state of Pt and weakened CMSI.

**The activation of C-H bond over the $Pt_1$-O($H^+$)-Fe.** Since the activation of the first C-H bond is regarded as the rate-limiting step of $CH_4$ combustion over noble-metal supported catalysts[67], $CH_4$-pulse reaction was conducted on the $Pt_1$/CF and $Pt_1$/CF-W SACs to investigate the water-treated impact on the C-H bond activation. As displayed in Table S6 and Fig. 5a, the consumption of $CH_4$ is increased 16-fold companied with more than 55 times $CO_2$ production on $Pt_1$/CF-W compared with that on the as-prepared $Pt_1$/CF, which suggests that not only the activation ability of Pt towards C-H bond is enhanced but also more active lattice O participate into the reactions after water treatment. It also confirms the Mars van Krevelen mechanism that differed from the normal reaction pathway reported on Pt nanocatalysts[68]. In the $CH_4$-TPR experiment, $Pt_1$/CF-W exhibited much lower temperature (~200 °C) for $CH_4$ consumption accompanied by the producing of $CO_2$ and larger yield of hydrogen and alkanes above the temperature of 250 °C as the consequence of methane directly pyrolysis compared with $Pt_1$/CF (Fig. S26). Thus, the enhancement of activity should be closely related to the improved $CH_4$ activation.

The key conclusion derived from above is that the function of water is turning the vicinal environment of Pt single atom by forming $Pt_1$-O($H^+$)-Fe structural unit. Therefore, an extrapolation should be viewed on other Fe-containing oxide-supported Pt catalysts where strong CMSI exists[25]. Indeed, in systems employing Fe-containing oxides such as $NiFe_2O_4$ and $Fe_2O_3$ as support, water promotion effect is successfully observed (Fig. 4b), while it is nearly absent in system where $MgAl_2O_4$ was served as a support (denoted as Pt/MA, Fig. S27).

In summary, a significant improvement of activity in $CH_4$ combustion by 50-fold is found in $Pt_1$/CF SACs after a simple

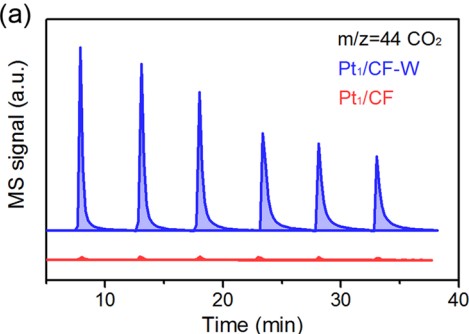
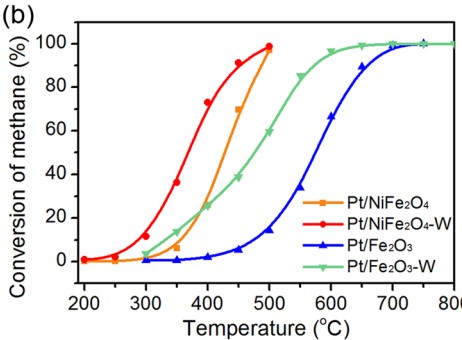

**Fig. 5 The activation of C-H bond. a** The generation of $CO_2$ on the $Pt_1$/CF and $Pt_1$/CF-W when injected $CH_4$ at 250 °C. **b** Light-off curves of $CH_4$ combustion over Pt/$NiFe_2O_4$ and Pt/$Fe_2O_3$ before and after water treatment catalysts. Reaction conditions: 0.5 vol% $CH_4$, 20 vol% $O_2$, $N_2$; GHSV = 40,000 ml/($g_{cat}$·h).

water-soaking treatment. More detailed study in both experiments and simulation reveal that it is not due to the direct participation of water in reaction but the modulation of the vicinal environment of Pt single atoms by $H^+$ that finally induced a weakened CMSI between Pt and $CoFe_2O_4$ spinel, and increase of catalytic activity. Extrapolation experiments are designed based on the physical picture we depict and show good universality of $H^+$-modulation across similar systems. This work provides a new strategy to promote the activity of SACs and may inspire new development in activity manipulation by tuning MSI through tailoring the vicinal environmental structure of metal atoms.

## Methods

**Materials**. The $CoFe_2O_4$ (CF) oxides were purchased from Aladdin and calcined at 500 °C for 4 h before use. To further purify the $CoFe_2O_4$, 2 g oxide was washed with 3 L ultrapure water, dried at 80 °C overnight and calcined in air, which is marked as $CF_p$. The ultra-pure water was produced on Millipore S.A.S. 67120 Molsheim with the 18.2 MΩ·cm$^{-1}$ resistivity at 25 °C. The $^{18}$O-labeled water ($H_2^{18}$O) was purchased from Aladdin with 97 atom % in $^{18}$O. $H_2PtCl_6$·$6H_2$O (99.99%) was purchased from Tianjin Fengchuan and platinum bis(acetylacetonate) (98%) was purchased from J&K Scientific company. All feed gases were provided by Dalian Special Gases Company.

**Synthesis of $Pt_1$/CF SAC and Pt/CF cluster catalyst**. The Pt/$CoFe_2O_4$ SACs with Pt loading of 1 wt% and 0.5 wt%, respectively, were synthesized by incipient wetness impregnation method. Typically, 2.0 g support was quickly added into 2.2 mL $H_2PtCl_6$ aqueous solution containing certain amount of Pt at room temperature. After impregnation for 24 h, the samples were dried at 60 °C overnight and calcined at 500 °C for 5 h (with a heating rate of 2 °C/min), marked as $Pt_1$/CF. Pt/$CoFe_2O_4$ cluster catalyst (denoted as Pt/CF) was synthesized with identical procedures except the loading of Pt increased to 2 wt%.

**Water treatment process**. Two types of water treatment processes were performed. (1) 0.1 g above catalysts were soaked with 100 mL ultra-pure water (18.2 MΩ·cm$^{-1}$) for 2 h, then filtered and dried in oven at 80 °C for 12 h. The obtained powder was labeled as $Pt_1$/CF-W. If there is no special statement, the water-treatment process was proceeded in this way. (2) To avoid the potential removal of any ions such as Cl$^-$, only 1 mL ultra-pure water was used and the suspension can be directly dried in oven at 80 °C for 12 h. The obtained powder was labeled as $Pt_1$/CF-W*.

**Synthesis of Pt(Ac)/CF**. Pt(Ac)/CF catalysts were synthesized using platinum bis(acetylacetonate) dissolved in ethanol. Ethanol was evaporated in a rotary evaporator and the resulting solid were dried at 60 °C and calcined at 500 °C for 2 h, respectively, marked as Pt(Ac)/CF.

**Synthesis of Pt/MA, Pt/$Fe_2O_3$, Pt/$NiFe_2O_4$**. The Pt/$MgAl_2O_4$, Pt/$Fe_2O_3$ and Pt/$NiFe_2O_4$ were prepared following the above impregnation procedure for Pt/CF except that home-made $MgAl_2O_4$, commercial $Fe_2O_3$ and $NiFe_2O_4$ were used as support. The metal loading of these catalysts is nearly 2.0 wt%.

**Synthesis of Pd/$Al_2O_3$**. Pd/$Al_2O_3$ catalysts were synthesized by impregnating the $Al_2O_3$ with an acetone solution of Pd($CH_3COO$)$_2$ (99.9%, Aladdin) with nominal weight loading of 1%. Excess acetone was removed in a rotary evaporator at 30 °C.

The resulting powder was then calcined at 500 °C under ambient air for 2 h and reduced at 500 °C in 10 vol% $H_2$/$N_2$ for 2 h forming fresh catalyst.

**Characterization**. The actual loadings of Pt were determined by inductively coupled plasma optical emission spectrometer (ICP-OES) on an ICP-OES 7300DV (PerkinElmer). The samples of Pt/CF can be dissolved completely in the hot fresh aqua regia. The microelements were further analyzed by the inductively coupled plasma-mass spectrometer (ICP-MS) on NexION 300D.

Scanning transmission electron microscopy (STEM) analyses were performed using a JEOL JEM-2100F operated at 200 keV. This instrument is equipped with a high-angle annular dark field scanning transmission electron microscopy (HAADF-STEM) detector and the resolution is 0.2 nm. Aberration-corrected HAADF-STEM images were obtained on a JEOL JEM-ARM200F STEM/TEM with a guaranteed resolution of 0.08 nm.

Temperature-programmed reduction by $H_2$ ($H_2$-TPR) and $CH_4$ ($CH_4$-TPR), $CH_4$-pulse reaction experiments, and CO chemisorption experiment were conducted on Autochem II 2920 with a thermal conductivity detector and a mass spectrometer. Prior to $H_2$-TPR, all samples were pretreated in flowing air at 120 °C for 0.5 h, and then cooled to −50 °C in argon. 70 mg samples were heated to 800 °C at a rate of 10 °C min$^{-1}$ in 10 vol% $H_2$/Ar with a flow rate of 30 mL/min. Before CO chemisorption, all samples were pretreated in Helium at 120 °C for 0.5 h, and then 5% CO/He was injected at 20 °C. For $CH_4$-pulse reaction, 70 mg samples were heated to 250 °C under He and purged for 10 min before dosing 20 vol% $CH_4$/He. For $CH_4$-TPR, the samples were purged with He for 10 min before heated to 500 °C at a rate of 10 °C min$^{-1}$ in 1 vol% $CH_4$/He. The content of $CH_4$, $CO_2$, CO and other hydrocarbons in the effluent gas were monitored by mass spectrometer.

Diffuse reflectance infrared Fourier transform spectra of CO adsorption (CO-DRIFTS) were acquired with a BRUKER Equinox 55 spectrometer equipped with a MCT detector at a spectral resolution of 4 cm$^{-1}$ and accumulation of 64 scans. Before each experiment, the sample was heated to 120 °C in He and maintained for 0.5 h, then the system was cooled down to room tempeature (20 °C). After the sample was purged with He for 1 h, the background was collected before introducing 3 vol% CO/He. Helium was then switched on to remove the gaseous CO and the spectrum was recorded simultaneously.

X-ray photoelectron spectroscopy (XPS) were performed on Thermofisher ESCALAB 250Xi instrument, equipped with monochromated Al Kα radiation as the X-ray source. The binding energies were referenced by setting the C 1 s binding energy to 284.8 eV. The surface of $Pt_1$/CF and $Pt_1$/CF-W were sputtered simultaneously with Ar$^+$ ions of 1000 eV at a low ion flux (0.2 μA·cm$^{-2}$). Ion scattering spectroscopy (ISS) was acquired using 800 eV He$^+$ ions and 80 eV analyzer pass energy.

The X-ray absorption fine structure (XAFS) at Pt $L_3$-edge of the samples were measured both at BL08B2[69] of SPring-8 (8 GeV, 100 mA) in Japan and BL14W of Shanghai Synchrotron Radiation Facility (SSRF) in China. The output beam was selected by Si (111) monochromator, and the energy was calibrated by Pt foil. The data of $Pt_1$/CF and $Pt_1$/CF-W were collected at room temperature with fluorescence mode. The spectra were analyzed and fitted using an analysis program Demeter[70]. The wavelet simulation was carried out with continuous Cauchy wavelet transform method[71].

**Isotope labeled experiments**. The isotope labeled experiments were carried out in the fixed-bed reactor connected with an online mass spectrometer. The m/z = 44 ($C^{16}O_2$), m/z = 46 ($C^{16}O^{18}O$), m/z = 48 ($C^{18}O^{18}O$) were recorded online. The $Pt_1$/CF catalyst (100 mg) was soaked with 1 ml $H_2^{18}$O and dried at 80 °C in vacuum for 12 h. After that, 100 mg catalyst was heated to 350 °C in the Helium gas flow and the temperature was remained at 350 °C at which the system was purged for 10 min until the system was stable. Then, the feed gas of 0.5 vol% $CH_4$ and 20 vol% $O_2$ balanced with $N_2$ was injected into the system and maintained for 10 min before cutting off.

**Catalytic oxidation of CH₄.** CH₄ oxidation reactions were performed in a U-shaped quartz fixed-bed reactor under atmospheric pressure. In a typical light-off experiment, 100 mg catalyst diluted with 2.0 g silica was used and the feed gas, 0.5 vol% $CH_4$, 20 vol% $O_2$ and $N_2$, was introduced with a flow rate of 33.4 ml/min. The temperature of the catalyst was measured with a k-type thermocouple inserted inside the reactor, touching the catalytic bed. The compositions of effluent gases were monitored online using a gas chromatograph (GC, Agilent 7890B) equipped with a thermal conductivity detector (TCD) and a flame ionization detector (FID). In kinetic experiments, gas hourly space velocity (GHSV) values were increased to control the conversion below 15% at a given temperature.

**DFT calculation.** All simulations were performed using the CP2K package[72]. Unrestricted Kohn-Sham formulation has been used as the electronic structure description method. Multigrid method is used to expand wavefunctions of valence treated electrons with Gaussian functions (DZVP-MOLOPT-GTH-SR) and plane waves, whose cutoff values are set to 50 Ry and 600 Ry, respectively[73,74]. For the description of core electrons, Goedecker-Teter-Hutter (GTH) pseudopotentials were used[75,76]. The PBE functional[77] with Grimme's D3 correction[78] was used to describe exchange-correlation and dispersion effect. Since classical generalized gradient approximated functionals always produce over-delocalized electron distributions, and failed to reproduce parts of electron correlation that was significant in Fe and Co elements, Hubbard corrections[79] for 3d electrons of Fe and Co were adopted. In this work, according to reported data for 3d transition metal oxides[80–82], a U-J ($U_{eff}$) value of 5.9 eV, ramping 1.0 eV was used, which provided a satisfying overall description of the electronic structure and surface reactivity[81]. Note that $CoFe_2O_4$ had ferrimagnetic phase below Curie point (~650 K), symmetry broken wavefunction was used for initial guess, where magnetization of Co and Fe atoms were set with opposite orientation. Surface dipole correction is applied in z-direction. Wavefunctions were optimized with Orbital Transformation (OT) method, whose convergence threshold was set to 1E-6 throughout calculations, Conjugated Gradient (CG) or Direct Inversion of the Iterative Subspace (DIIS) minimizer was used depending on difficulty to get convergence. Structures were optimized using the Broyden–Fletcher–Goldfarb–Shannon (BFGS) or CG algorithm, and default convergence thresholds were used (maximum geometry change <3.0E-3 bohr; maximum force component <4.5E-4 hartree/bohr; RMS of maximum geometry change <1.5E-3; RMS of maximum force component <3.0E-4 hartree/bohr). Details about structures and tests are provided in Supplementary information.

## Data availability

All data within the article and its supplementary information file are available from the authors upon request.

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

## Acknowledgements

This work was financially supported by National Natural Science Foundation of China (21972135, 21961142006, 21776270, 21776271, 51701201), CAS Project for Young Scientists in Basic Research (YSBR-022), the Strategic Priority Research Program of the Chinese Academy of Sciences (XDA21040200), and the DNL Cooperation fund, CAS(DNL201903). The authors thank the approval of Japan Synchrotron Radiation Research Institute (Proposal Nos. 2019B3415, 2020A3415) and the beamline staff both at BL08B2 of SPring-8 and at BL14W of SSRF.

## Author contributions

J.Y. prepared the catalysts, performed the reaction tests and most of the characterizations. Y.H. did the theoretical calculation and analysis. H.Q. helped to perform a part of characterizations and revise the manuscript. C.Z. and Q.J. performed the aberration-corrected electron microscopy characterization. Y.C. helped to perform the synchrotron radiation experiment in Japan. Y.S. and X.P. helped to do the electron microscopy characterization. X.D. helped to perform some experiments. X.L. and A.W. provided suggestion on the XAFS measurement and data fitting. W.L., B.Q. and T.Z. conceived the idea and directed the project. J.Y. and B.Q. co-wrote the manuscript. All the authors discussed the results and commented on the manuscript.

## Competing interests

The authors declare no competing interests.
