## [Peer Review File · Nature Communications]

Title: Modulating the strong metal-support interaction of single-atom catalysts via vicinal structure decorationREVIEWER COMMENTS

Reviewer #1 (Remarks to the Author):

This paper represents very interesting results on CH₄ combustion over supported Pt single-atom catalyst. The water soaking treatment of Pt/CoFe₂O₄ for 2h significantly lowered T₅₀ temperature of CH₄ combustion from 468 °C to 325 °C. The light-off temperature was around 250 °C. This is amazing promotion because Pt is known to be not preferable for CH₄ combustion. This paper is excellent from an engineering point of view. While, from a scientific point of view, the authors do not yet solve why the soaking enhanced the catalytic activity. I recommend the major revision on the following points.

- Figure of 2a (CO-DRIFT) and Figure 2c (H₂-TPR) show the increases in CO adsorption sites and the number of redox sites. The results suggest a big structural arrangement of the catalyst surface. These results indicate that the promotion effect cannot be rationalized by the local rearrangement and contribution of H⁺ as authors mentioned. I strongly recommend the discussion based on the quantification of CO adsorption sites and redox sites.
- FeO_x and CoO_x are slightly soluble in water. The big rearrangement may cause by the solved and deposited Fe or Co. XPS analysis of Fe, Co, and O may help their discussion.
- The quantification of the pulse CH₄ analysis (Figure 5a) also help their discussion.

Reviewer #2 (Remarks to the Author):

The manuscript by Yang and coworkers reports their findings of catalytic activity promotion after water treatment of their Pt/CoFe₂O₄ single atom catalysts (SACs) for CH₄ combustion. They found that the catalytic activity of their Pt/CF SACs for CH₄ combustion increased by more than 50 folds at 300 °C by simply soaking Pt/CF SAC in water. They attributed the observed activity promotion to possible vicinal structure decoration due to H⁺ which changes the metal-support interaction between Pt and CoFe₂O₄ support. While the observed activity promotion of water treatment is impressive, the interpretation and the conclusions draw from this paper are not well-supported. In addition, the writing of the manuscript need to be polished. Therefore, I recommend a major revision of the current manuscript.

Reviewer #3 (Remarks to the Author):

The paper reports on the CMSI of Pt/CoFe₂O₄ weakened by simple water soaking treatment has a positive effect on the C-H bond activation during the methane combustion reaction. The authors devised a method to maximize the effect of a single atom with a simple treatment method, and this was proven through various characterizations. However, the direct reason why small structural differences, such as the decoration of H⁺ at the Pt-O-Fe interface, actually boosting the activity significantly, is not clearly explained analytically. The authors should report the apparent activation energies and longevity tests for the catalysts to show whether this difference activity is a transient phenomenon.

Point-by-point response to reviewers' comments

Manuscript ID: NCOMMS-21-43244

Title: Modulating the strong metal-support interaction of single-atom catalysts via vicinal structure decoration

Reviewer #1

“This paper represents very interesting results on CH₄ combustion over supported Pt single-atom catalyst. The water soaking treatment of Pt/CoFe₂O₄ for 2h significantly lowered T₅₀ temperature of CH₄ combustion from 468 °C to 325 °C. The light-off temperature was around 250 °C. This is amazing promotion because Pt is known to be not preferable for CH₄ combustion. This paper is excellent from an engineering point of view. While, from a scientific point of view, the authors do not yet solve why the soaking enhanced the catalytic activity. I recommend the major revision on the following points.”

We sincerely thank the reviewer for his/her valuable comments and for acknowledging the importance of this manuscript. Specific responses and discussions in details are listed below.

“1. Figure of 2a (CO-DRIFT) and Figure 2c (H₂-TPR) show the increases in CO adsorption sites and the number of redox sites. The results suggest a big structural arrangement of the catalyst surface. These results indicate that the promotion effect cannot be rationalized by the local rearrangement and contribution of H⁺ as authors mentioned. I strongly recommend the discussion based on the quantification of CO adsorption sites and redox sites.”

Response:

We sincerely thank the reviewer for these good comments with which, however, we have to respectively disagree. Firstly, the difference in CO adsorption indeed suggested a change of adsorption property of the catalyst (or more accurately, the supported Pt atoms). However, this doesn't necessarily mean a big structure change of Pt, especially regarding to the dispersion (size) of Pt. We totally understand the

reviewer's concern that for traditional supported nanocatalysts without the presence of strong metal-support interaction (SMSI), CO adsorption is usually used to measure the dispersion of Pt-group metals (PGMs) and the change of CO adsorption amount does reflect the changes in dispersion. However, it is another story in catalyst systems with strong metal-support interactions, and the CO adsorption amount cannot be simply correlated to the dispersion (size) of PGMs due to the presence of other effects such as the electronic effect (*Nat. Commun.*, 2019, 10, 234) or the physical coverage of PGM sites by oxide layer (*Nat. Commun.*, 2020, 11, 5811). For example, in our recent work, the Pt/TiO₂ SAC loses its ability to adsorb small molecular upon high-temperature reduction because of the coordination saturation rather than dispersion change (*Angew. Chem. Int. Ed.*, 2020, 59: 11824-11829). Similarly, in the current work, only 2 $\mu\text{mol/g}$ CO was adsorbed on Pt₁/CF measured by CO chemisorption experiment, consistent with the tiny peak observed in CO-DRIFTS and presenting good evidence of the covalent metal-support interaction (CMSI, *Nat. Commun.*, 2019, 10, 234). After water treatment, the CO adsorption amount increased significantly to about 19 $\mu\text{mol/g}$ which is still much lower than the theoretical value of ~ 50 $\mu\text{mol/g}$ (**Table R1**). Therefore, the CO adsorption capacity in this work cannot reflect the dispersion of Pt but only a different degree of CMSI. To further release the review's concern, we have carried out ion scattering spectroscopy (ISS, which is sensitive to only one atom layer) to detect the possible quantitative change of Pt single atoms on the outermost surface. As shown in **Figure R1**, the Pt:Fe atomic ratio in the outermost surface layer keep almost constant, clearly illustrating that the number of single atoms on the surface remained unchanged.

Table R1. The quantity of CO adsorption on the Pt₁/CF and Pt₁/CF-W.

Sample	Metal loading (wt %)	CO adsorption quantity ($\mu\text{mol/g}$)	
		Theoretical	Actual
Pt ₁ /CF	1.0	51.3	2.28
Pt ₁ /CF-W	1.0	51.3	19.3

Figure R1. ISS spectra of Pt₁/CF and Pt₁/CF-W.

On the other hand, the quantitative analysis of H₂-TPR showed that the consumption of H₂ in both cases are much higher than the theoretical one, suggesting the partial reduction of support stemmed from the H₂ spillover. Thus the quantification of redox site based on H₂-TPR is also inaccurate. It is notable that the temperature of H₂ consumption was decreased accompanied with a slightly increased amount on Pt₁/CF-W sample (**Table R2**), which might be related to the enhanced ability of Pt towards H₂ activation with lower valence and/or the improved reducibility of support after H⁺ modification. Therefore, the H₂-TPR result indeed suggests a change of redox property rather than a big structure change.

Table R2. The quantity of H₂ consumption on the Pt₁/CF and Pt₁/CF-W.

Sample	Metal loading (wt %)	H ₂ adsorption quantity (μmol/g)	
		Theoretical	Actual
Pt ₁ /CF	1.0	102	438
Pt ₁ /CF-W	1.0	102	537

We have added this new data in the supporting information (Figure S4, Table S1-2) and the corresponding discussion into the revised manuscript on page 4-5.

“2. FeO_x and CoO_x are slightly soluble in water. The big rearrangement may cause by the solved and deposited Fe or Co. XPS analysis of Fe, Co, and O may help their discussion.”

Response:

We thank the reviewer for his/her valuable comments. The K_{sp} of $Fe(OH)_3$ and $Co(OH)_2$ are 1.8×10^{-36} and 1.9×10^{-15} , respectively (*Soil Syst.*, 2018, 2(2): 20), which means they are very hard to be dissolved in the water, not to mention their oxide counterparts. To convince the reviewer, we soaked 0.5 g Pt₁/CF (25.6 μ mol Pt) catalysts in 5 mL water and examined the ion concentration of the filtered solution by ICP-MS. No Fe ion was detected, suggesting the concentration of Fe ions was below detection limit. The Co concentration was around 1 ppm, which means that 0.08 μ mol Co was dissolved. It's hard to believe that such a low concentration of Fe/Co dissolution in water, a few orders of magnitudes lower than the quantity of Pt, would have arouse huge structure change.

In addition, the possible changes on surface O/Fe/Co properties were, according to the reviewer's suggestion, examined by XPS analysis. The O1s spectra show that no detectable changes of adsorbed water (533 eV) and lattice oxygen (529.7 eV) were observed on Pt/CF-W compared with Pt/CF (**Figure 4b**). It is obvious that the quantity of OH⁻ group (531.3 eV) on Pt/CF-W is more than that on Pt/CF, which must have come from the dissociation of water. The shift of both Fe 2p and Co 2p spectra to higher energy together with a detailed fitting analysis according to Biesinger et al (*Appl. Surf. Sci.*, 2011, 257(7): 2717-2730) further confirmed the hydroxylation of FeO_x or CoO_x by water (**Figure R2, Table R3**).

Table R3. XPS analysis of the Pt, O, Fe and Co properties on the surface of samples.

Sample	O (%)			Fe (%)		Co (%)	
	H ₂ O	Fe/Co-OH	Lattice O	CoFe ₂ O ₄	FeOOH	CoFe ₂ O ₄	Co(OH) ₂
CoFe ₂ O ₄	-	0.12	0.88	~1	~99	~100	0
Pt/CF	-	0.13	0.87	~100	0	~99	~1
Pt/CF-W	-	0.28	0.72	~95	~5	~86	~14

Figure R2. XPS spectra of (a) Fe 2p (b) Co 2p and (b) O 1s observed on CoFe₂O₄, Pt₁/CF and Pt₁/CF-W.

We have added the XPS of Fe, Co and O data into the supporting information in Figure S23 and corresponding discussion in the revised manuscript on page 9.

“3. The quantification of the pulse CH₄ analysis (Figure 5a) also help their discussion.”

Response:

We thank the reviewer’s suggestion. As suggested by the reviewer, we have qualification the consumption of CH₄ in the pulse experiment.

Table R4. The quantification of CH₄ and CO₂ in the CH₄-pulse experiment.

Pulse	The consumption of CH ₄ (μmol/g _{cat})		The production of CO ₂ (μmol/g _{cat})	
	Pt ₁ /CF	Pt ₁ /CF-W	Pt ₁ /CF	Pt ₁ /CF-W
1	3.49	39.18	0.16	12.35
2	1.85	32.39	0.24	12.83
3	2.76	32.36	0.24	12.07
4	2.17	35.23	0.26	11.32
5	1.80	35.48	0.20	10.31
6	1.42	33.45	0.17	9.50

From the **Table R4**, we can see that the consumption of CH₄ is increased 16 folds after water treatment companied with more than 55 times CO₂ production, which may

stem from different degrees of coke formation on the two samples during CH₄ pulse and suggests that not only the activation ability of Pt towards C-H bond is enhanced but also more active lattice O participate into the reactions after water treatment. Based on our DFT calculation, when the water dissociated adsorption on the surface of Pt₁/CF, the valence state of Pt slightly decreased, which contributed to the activation of C-H bond (*J. Phy. Chem. C*, 2011, 115: 944-951; *J. Am. Chem. Soc.*, 2011, 133(40): 15958-15978), and the strength of Pt-O bond was weakened thus the lattice O was easier to take off.

We have added the **Table R4** in the supporting information and the corresponding discussion in the revised manuscript on page 11.

Reviewer #2

“The manuscript by Yang and coworkers reports their findings of catalytic activity promotion after water treatment of their Pt/CoFe₂O₄ single atom catalysts (SACs) for CH₄ combustion. They found that the catalytic activity of their Pt/CF SACs for CH₄ combustion increased by more than 50 folds at 300 °C by simply soaking Pt/CF SAC in water. They attributed the observed activity promotion to possible vicinal structure decoration due to H⁺ which changes the metal-support interaction between Pt and CoFe₂O₄ support. While the observed activity promotion of water treatment is impressive, the interpretation and the conclusions draw from this paper are not well-supported. In addition, the writing of the manuscript need to be polished. Therefore, I recommend a major revision of the current manuscript.”

“1. Although the authors claim they have made Pt/CoFe₂O₄ single atom catalysts, the uniformity of the Pt SACs on support is questionable. There are CO-DRIFT peaks in the range of 1800-1900 cm⁻¹ in Figure 2a, which is in the region for CO bridged adsorption. Their intensity decreases after water-treatment. Therefore, a possible can be as follows: Pt₁/CF contains Pt SAC and Pt small cluster; in Pt₁/CF-W, small Pt clusters are further dispersed to form more Pt SACs. Thus, the IR intensity of bridge-binding CO (in the range of 1800-1900 cm⁻¹) decrease. In fact, in the STEM images provided in the paper, there are bright points next to each other which could be small Pt clusters.”

Response:

We sincerely thank the reviewer for raising these set of questions. We would like to clarify that the peaks in the region of 1800-1900 cm⁻¹ are in fact noise signals rather than CO adsorption peaks. This conclusion was obtained based on the following two aspects. First of all, the bridged CO adsorption on Pt is a single peak (or broad band) centered at ~1860 cm⁻¹ while the signals located in the region of 1800-1900 cm⁻¹ (actually 1700-2000 cm⁻¹) in this work were numerous and irregular, which are more likely to be noise caused by moisture in the light path outside the cell. In addition, the

peak position of CO linear adsorption is independent of the CO coverage, an obvious character of the single-dispersed Pt species (Figure S3, Chem. Rev., 2020, 120: 11986-12043). Therefore, we can safely claim that there was no CO bridged adsorption on our samples. To further convince the reviewer, we have re-performed the CO-DRIFT experiment trying to exclude the effect of moisture by introduce dry N₂ gas around the chamber. It is clear that no signals were observed in the region of 1800-1900 cm⁻¹ **in the Figure R3**, unambiguously excluding the presence of CO bridged-adsorption peak. In addition, we believe it's hard, if not impossible, to re-disperse the small Pt clusters/nanoparticles into Pt single atom simply by water soaking. If this was the case, it might be a more striking and more valuable discovery for the preparation of SACs with simple treatment. However, no evidence had been found to support this possibility so far. And theoretically, detaching Pt atoms from Pt nanoparticles/crystal needs high energy input, for example, calcination at much higher temperatures (Science, 2016, 353: 150-154; *Nat. Commun.*, 2019, 10: 234; *Sci. China Mater.*, 2020, 63, 949-958; *Nat Commun.*, 2020, 11: 1263). The STEM image indicated the prevalence of isolated Pt atoms and the bright points are more likely to be neighboring Pt single atoms considering that the high density of Pt single atoms on the CoFe₂O₄ with 1.0 Pt/nm² (theoretical value). We have replaced the old CO-DRIFT data with the new one in our revised manuscript.

Figure R3. (a) CO-DRIFT spectra of Pt₁/CF-repeat, Pt₁/CF, Pt₁/CF-W and (b) Time-resolved CO-DRIFT spectra when purging the CO with He.

“2. Why does 1 wt% Pd/Al₂O₃ result in Pd nanoparticle of 1.4 nm and 1 wt% Pt/CoFe₂O₄ result in Pt SACs? Can it be that there is a mixture of Pt SACs and small Pt clusters in 1 wt% Pt/CoFe₂O₄, as suggested by CO DRIFTS and HAADF-STEM. The authors showed that 2 wt% Pt/CoFe₂O₄ resulted in Pt NPs. A nature question is that why 1 wt% Pt/CoFe₂O₄ result in Pt SACs? Given the poor quality of the HAADF-STEM images and possible clusters of bright points in those images, as well as the CO peaks in the range 1800-1900 cm⁻¹, control experiments with lower loading of Pt need to be performed to support the so-called Pt₁/CF SACs is indeed predominantly Pt SACs and the activity promotion due to water treatment is still observed for lower loading of Pt.”

Response:

Thank you for these series of questions. The dispersion state of the metal species is closely related to the metal-support interaction (MSI) which is in turn determined by the nature of the metals and supports. Our previous studies have revealed that the covalent metal-support interaction (CMSI) between Pt-group metals (PGMs) and Fe-containing oxides can help to re-disperse the PGM nanoparticles into single atoms through high-temperature calcination while the PGM nanoparticles sintered on Al₂O₃ upon high-temperature calcination (*Nat. Commun.*, 2019, 10: 234; *Sci. China Mater.*, 2020, 63, 949-958, *Nat Commun.*, 2020, 11: 1263). One of the features/advantages of this finding for SAC preparation is that the metal loading mainly depends on the surface area rather than the density of vacancy of the support (the former is often much higher than the latter), offering a new strategy to prepare thermally stable SACs with high metal loadings. In current work, the BET surface area of CoFe₂O₄ is 28.4 m²/g, corresponding to a maximum metal loading of 1.5 wt%. Therefore, it's reasonable that 1 wt% Pt₁/CF is SAC while the 2 wt% Pt/CF contains some small clusters/nanoparticles. As mentioned above, the poor quality of the CO-DRIFT measurement was due to the high humidity. We have now provided a new result with much better quality. Therefore, we can safely expect that all Pt atoms will be singly dispersed if the Pt loading is further lowered according to the reviewer's suggestion. To convince the reviewer, we have prepared a 0.5 wt% Pt₁/CF catalyst and the

AC-HAADF-STEM images shown in the **Figure R4** demonstrate that the Pt metals are indeed fully dispersed as single atoms on the CoFe_2O_4 . We also examined the water-treatment effect on the catalytic performance of 0.5 wt% Pt_1/CF in the CH_4 -TPR and light-off experiment. As shown in **Figure R5a**, the maximum consumption of CH_4 is located at 320 °C, while the temperature sharply decreases to 284 °C on the water-treated $\text{Pt}_1/\text{CF-W}$. Significant activity increase was also observed in the light-off experiment (**Figure R5b**), which illustrated the activation of CH_4 was enhanced on the water-treated SAC.

Figure R4. AC-HAADF-STEM images of (a) 0.5 wt% Pt_1/CF and (b) $\text{Pt}_1/\text{CF-W}$.

Figure R5. (a) CH_4 -TPR profiles of 0.5 wt% Pt_1/CF and $\text{Pt}_1/\text{CF-W}$ under 1 vol% CH_4 , 99 vol% He; (b) light-off curves of methane combustion of 0.5 wt% Pt_1/CF and $\text{Pt}_1/\text{CF-W}$.

Reaction conditions: 0.5 vol% CH_4 , 20 vol% O_2 , N_2 ; GHSV = 20,000 ml/($\text{g}_{\text{cat}} \cdot \text{h}$)

In the revised manuscript, we have supplemented the results of 0.5 wt% Pt₁/CF on the page 9, and explained the formation mechanism of such high-density Pt single atoms on CoFe₂O₄ on the page 3. The Figure R4 and R5 were added in the supporting information as Figure S19 and S20.

“3. Regarding the size effect, the authors mentioned one possibility that the water treatment will not induce the aggregation between Pt SAC. However, the possibility that water treatment may induce the disperse of small Pt cluster in the currently denoted Pt₁/CF to more single-atom dispersed Pt SACs in Pt₁/CF is not considered. There possibility is quite likely considering that the CO IR peaks in the range of 1800-1900 cm⁻¹ decreases by water treatment. From Figures S17, we can see that the activation for CH₄ combustion increases when we make Pt more dispersed in Pt₁/CF SAC than Pt/CF cluster. A good control experiment could be done is to prepare Pt₁/CF SAC with wt% lower than 1%, if the specific activity per Pt is more or less similar to that of Pt₁/CF SAC with 1 wt%. There is no guarantee that the denoted Pt₁/CF SAC is indeed uniformly Pt SAC as claimed in the paper, especially considering the IR peaks in Figure 2a.”

Response:

Table R5. The performances of Pt-based catalysts for CH₄ combustion.

Sample	Pt (wt %)	GHSV ml/(g·h)	Reaction rate* (mmol mol _{metal} ⁻¹ s ⁻¹)
Pt ₁ /CF	1.0	20,000	0.3
Pt ₁ /CF	0.5	20,000	0.5
Pt ₁ /CF-W	1.0	80,000	16.3
Pt ₁ /CF-W	0.5	40,000	14.8

Thank you for these questions/comments which have actually been included in your question 1 and 2 and we have answered them in the above response and here we just summarized briefly: 1) the CO-DRIFT was due to noise signals of moisture and we have now provided updated ones. 2) The redispersion of Pt small clusters/nanoparticles seems impossible merely by simple water treatment as high

energy input is required to detach the Pt single atoms from particles/clusters. 3) We have prepared Pt₁/CF with lower metal loading (0.5 wt% Pt₁/CF) and measured their specific rate before and after water treatment, which are all similar to that of 1.0 wt% Pt₁/CF-W (**Table R5**).

We have added the data into Table S3 in the supporting information.

“4. The claim that “some literature suggested that the aggregation of Pt single atoms gave rise to better activity.” The authors cited ref. 58, which does not provide support to this claim.”

Response:

We sincerely thank the reviewer for raising this question. In ref. 58, the activity increased when the single atoms evolved into metal clusters and nanoparticles in CO oxidation, dehydrogenation of propane and photocatalytic H₂ evolution reactions. As for methane combustion, we have supplemented another literature where the activity of Pt/CeO₂ cluster is higher than that of Pt₁/CeO₂ SAC (*Nat. Catal.*, 2020, 3, 824-833).

“5. The Pt 4f XPS suggest there are ~76% Pt 4+ and ~24% Pt 2+ in Pt₁/CF and ~68% Pt 4+ and ~32% Pt 2+ in Pt₁/CF-W. What is the error bar of the atom percentage in those estimates? Do the numbers is reliable enough to indicate the chemical states of Pt change after the water treatment? If so, what could be the reason cause the partial reduction of Pt simply by soaking Pt₁/CF in water? As mentioned before, the intensity of IR peaks of bridged-CO in Pt₁/CF in the range of 1800-1900 cm⁻¹ decreases after water treatment (Figure 2a).”

Response:

This is a very good question. In fact, the conclusion that Pt chemical state change after water treatment was mainly obtained from the CO-DRIFT results (the linear CO adsorption has a red-shift). The XPS as well as XANES was used to further verify this conclusion and same trends were obtained. Therefore, although we totally agree with the reviewer that these values from XPS might not be very accurate and might be

influenced by the radiation of X-ray and photoelectron produced, we believe the conclusion is reliable. Actually, to avoid this effect as far as possible, we have rigorously kept the same test conditions in our original test. In addition, repeatable measurement of another bench of catalysts verifies our conjecture (**Figure R6**).

Figure R6. Repeated Pt 4f XPS spectra

To fundamentally answer the question why simply water soaking can cause reduction of Pt, Charge Density Difference (CDD) was estimated for structure of water-soaked catalyst modelled in DFT simulation (recalculated, shown in **Figure R7**). Before interpreting CDD result, it needs to revisit some knowledge of DFT: after adsorption of proton H^+ and OH^- group from spontaneous water dissociation, charges re-distribute according to atoms' intrinsic properties such as electronegativity χ and chemical hardness η , (gradient and curvature of energy respect to number of electrons, respectively), and from Hard-Soft-Acid-Base theory, number of electrons transferred can actually be determined by χ and η . For example, a two-atom system, atoms' energies can be expanded respect to number of electrons:

$$\begin{cases} E_1(N_1) = E_1(N_1^\ominus) - \chi_1(N_1 - N_1^\ominus) + \eta_1(N_1 - N_1^\ominus)^2 \\ E_2(N_2) = E_2(N_2^\ominus) - \chi_2(N_2 - N_2^\ominus) + \eta_2(N_2 - N_2^\ominus)^2 \end{cases}$$

where $E_1(N_1)$ and $E_2(N_2)$ denote energy of atom 1 and 2 when possessing N_1 and N_2 electrons respectively. N_1^\ominus and N_2^\ominus are electrons numbers of atom 1 and 2 before electron transfer. Electronegativity and chemical hardness are defined as:

$$\begin{cases} \chi_i = -\left(\frac{\partial E_i}{\partial N_i}\right)_v \\ \eta_i = \frac{1}{2}\left(\frac{\partial^2 E}{\partial N_i^2}\right)_v \end{cases}$$

Because total number of electrons will never change after reaction, conservation of number of electrons must hold: $N_1 + N_2 = N_1^\ominus + N_2^\ominus$. Denote number of electrons transferred ΔN as $N_1 - N_1^\ominus = -(N_2 - N_2^\ominus) \equiv \Delta N$, then total energy can be written as:

$$\begin{aligned} E &= E_1 + E_2 \\ &= E_1(N_1^\ominus) + E_2(N_2^\ominus) + (\chi_2 - \chi_1)\Delta N + (\eta_1 + \eta_2)\Delta N^2 \end{aligned}$$

To minimize total energy,

$$\Delta N = \frac{\chi_1 - \chi_2}{2(\eta_1 + \eta_2)}$$

, which indicates number of electrons transferred is determined by both electronegativities and chemical hardness of electron donor and acceptor.

Based on knowledge revisited above, it is straightforward that H^+ and OH^- group adsorption change charge distribution. With more details and as shown in the following **Figure R7**, atoms from water molecule are subscripted with “w”. Note that neutral O atom has electron configuration as $1s^2 2s^2 2p^6$, only another two electrons are needed. In most cases if O forms more than two covalent bonds, lone pair (LP) electrons must participate in bond formation, which further results in an electron-donation from LP. This is the reason for bond3 and 4 return a part of electrons to Pt. The O_w-H_w bond length is 0.9953 Angstrom, similar with O-H bond length in H_3O^+ ion, 0.9910 Angstrom (estimated with identical method and basis sets), implying O_w-H_w , O_w-Fe and O_w-Pt are mixture of classical covalent bond and coordination bond.

Figure R7. Charge density difference of water-soaked structure modelled in DFT simulation. CDD is calculated by $\Delta\rho = \rho_{\text{ads}} - \rho_{\text{unads}} - \rho_{\text{OH, H}}$, where ρ_{unads} is electron density of unrelaxed, non-adsorbed structure, $\rho_{\text{OH, H}}$ is electron density of unrelaxed dissociated water molecule (restricted Kohn-Sham and unrestricted Kohn-Sham with symmetry broken methods setting obtain identical reasonable result).

Mayer bond order analysis (*J. Comput. Chem.*, 2012, 33: 580-592) is carried out for relaxed structures both before and after water soaking:

Table R6. Mayer bond order of Pt-O bonds before and after water-soaking

Pt-O bond order	1	2	3	4
Before	0.7119	0.8305	1.1249	
After	0.6570	0.7499	0.6029	0.5646

The fourth Pt-O bond formed has the lowest bond order, is very easy to break upon molecule adsorption, such as CO adsorption (**Figure R8**)

Figure R8. Model of CO adsorption on water-soaked catalyst. Because difficulty in both converging wavefunction and geometric property of present system, structure is relaxed with cutoff of planewave basis decreases from 600 to 550 Ry, cutoff for real space gridding of Gaussian basis function decreases from 50 to 30 Ry.

“6. The H₂-TPR shifted to a distinctly lower temperature (144 to 90 degree) after water treatment? Why do the more reduced Pt SACs after water treatment, as claimed by the authors, have a lower temperature for H₂ reduction? From chemical intuition, it seems that more reduced Pt SACs should have a higher temperature for H₂ reduction since they have already been partially reduced. Can the authors explain this apparently inconsistency?”

Response:

In most Pt SAC systems reported so far, Pt single atoms are bonded to the support via strong electronic interactions and the resulting high valence state of Pt single atoms weakens their capability to activate hydrogen molecular (*J. Am. Chem. Soc.*, 2019, 141: 8185-8197, *Nat. Commun.*, 2021, 12: 3295). This is supported by the fact that SACs have usually lower hydrogenation activity despite much better selectivity (*Chem. Rev.*, 2020, 120: 683-733). Therefore, lowering the oxidation state of single metal atoms is beneficial to the activation of H₂ which was supported by numerous previous reports. For example, Ren and his coauthors discovered that the hydrogenation activity of Pt SAC (supported on Fe₂O₃) significantly increased due to the easier dissociation of H₂ on the reduced Pt single atoms (*Nat. Commun.*, 2019, 12, 3295). Therefore, the temperature for the H₂ reduction over more reduced Pt SACs should be lower than that of oxidized Pt SACs.

“7. Is the tiny difference of white line intensity of Pt₁/CF and Pt₁/CF-W (Figure S4) is reliable to draw the conclusion that the Pt centers in Pt₁/CF-W have a lower electronic state, especially considering the non-uniformity of the Pt dispersion (from CO-DRIFT (Figure 2a) and HAADFF-STEM (Figure 1b)). In addition, the white line intensity of PtO₂ is much lower than both Pt₁/CF and Pt₁/CF-W, suggesting the Pt in

PtO₂ has a lower chemical state following the authors' argument, which is clearly not the case. In the EXAFS analysis (Figures 2d and S5), the Pt-O-Fe distances actually fall between the Pt-Pt distances. It may be an overstatement that "in the detailed wavelet transform EXAFS (WT-EXAFS) analysis confirmed the sole presence of Pt single atoms", especially considering the presence of CO vibrational peaks in the range of 1800-1900 cm⁻¹. It is also possible that there are partially small Pt clusters present in the Pt₁/CF, while the water-treatment induces further disperse of Pt to have more actual Pt SACs in the Pt₁/CF-W samples."

Response:

We thank the reviewer for his/her valuable comments. As mentioned above, this question is based on the misunderstanding of our previous CO-DRIFT results. As mentioned above, the conclusion of chemical state change was mainly derived from the CO-DRIFT, and XPS and XANES results are complementary evidence to further verify our conclusion. We acknowledge that the difference of white line intensity of Pt₁/CF and Pt₁/CF-W seems too small to directly confirm the different electronic state of Pt. But we have also provided other characterization results, such as XPS and CO-DRIFTS, to verify the lower electronic state of Pt after water treatment. When all evidences point to one conclusion, we believe it is much more believable. As for why the white line intensity of Pt₁/CF and Pt₁/CF-W is higher than the PtO₂, we think it might be due to the strong interaction of Pt₁ single atoms with CoFe₂O₄ support which results in larger population of unoccupied Pt 5d states (*J. Am. Chem. Soc.*, 2019, 141(37):14515-14519) that has been frequently reported.

For the EXAFS data, because of the similar bond distance of Pt-O-Fe and Pt-O-Pt/Pt-Pt, it is difficult to make a clear judgment from Fourier-transformed k²-weighted EXAFS spectra (R space) only. Therefore, we have provided the WT-EXAFS analysis, which integrated the information of both K space and R space. From **Figure R9 (also Figure S6)**, we can distinctly differentiate the Pt-O-Fe and Pt-O-Pt/Pt-Pt because the lobe of the Pt-O-Fe at (2.5 Å, ~5.5 Å⁻¹) is far away from that of Pt-O-Pt (3.0 Å, >10.0 Å⁻¹) and Pt-Pt (2.5 Å, 10.0 Å⁻¹). Therefore, the

WT-EXAFS analysis has well illustrated the absence of Pt-Pt/Pt-O-Pt bond in our Pt₁/CF and Pt₁/CF-W SACs.

As for the CO-DRIFT data and possibility of re-dispersion of Pt small clusters, please refer to the response of the question 1.

Figure R9. Wavelet transform of EXAFS for Pt foil, PtO₂, Pt₁/CF and Pt₁/CF-W.

“8. In line 140 on page 6, experimental condition, ‘0.5 vol% CH₄, 20 vol% O₂ and N₂’, do the authors mean “0.5 vol% CH₄, 20 vol% O₂ and 79.5 vol% N₂’?”

Response:

Yes, you are right. Many thanks for pointing this out. We have modified the description in the experimental section and made it clearer that “the catalyst is evaluated under 0.5 vol% CH₄, 20 vol% O₂ and 79.5 vol% N₂.”

“9. In general, no error bars haven been provided in the experimental light-off curves. It maybe better to provide error bars of the data points in Figure 3a, S7, S8, etc.”

Response:

We sincerely thank this reviewer for raising this question. Actually, the repeatability of this series of catalysts in CH₄ combustion was quite good in our

repeated experiments. According the reviewer's suggestion, we have additionally repeated the light-off curves of all catalysts in all catalytic performance tests for four times based on which we provided the error bars as shown in the following **Figure R10**.

Figure R10. Repeated Light-off curves of CH₄ combustion over (a) Pt₁/CF-W, (b) Pt₁/CF catalysts. (c) The light-off curves of CH₄ combustion with an error bar.

The reaction conditions: 0.5 vol% CH₄, 20 vol% O₂, and 79.5 vol% N₂, GHSV = 20,000 ml/(g_{cat}·h).

We also provided error bar into Figure S7, Figure S8, Figure S14 and Figure S17 (as shown in **Figure R11**). These data have been added in the revised manuscript and supporting information to replace the old ones.

Figure R11. The light-off curves of CH₄ combustion with an error bar. (a) Pd/Al₂O₃; (b) CoFe₂O₄ and CoFe₂O₄-W; (c) Pt(Ac)/CF and Pt(Ac)/CF-W; (d) Pt/CF Cluster.

The reaction conditions: 0.5 vol% CH₄, 20 vol% O₂, and 79.5 vol% N₂, GHSV = 20,000 ml/(g_{cat}·h).

“10. The only direct support of H⁺ decoration comes from theoretical modelling which is not well performed. Details regarding the computational modeling such as spin-states, alignment of spins at Fe and Co centers, terminations of the CoFe₂O₄ (110) facets need to be provided and properly treated. In addition, coordination environments of Pt in experimental characterization (coordination number of Pt-O shell ~ 4) and theoretical modeling (coordination number of Pt-O shell = 3) are very different, raises question whether the theoretical modeling is relevant to the actual system.”

Response:

We sincerely appreciate reviewer for raising this issue. Because temperatures of reaction and calcination (500 °C) were close and far above Curie point respectively (~650 K, **Figure R12a**, *J. Magn. Magn. Mater.* 2009, 321(9): 1251-1255), at first supports in models we planned to construct were assumed to be paramagnetic, which means symmetry-breaking was not manually set in wavefunction optimization. We fully agree with you that spin-alignment is important for this kind of ferrimagnetic material, thus after re-consideration, we think because all our characterization have been performed at room temperature and our reaction temperature indeed has a non-negligible range below Curie point, symmetry broken system is a more appropriate choice for discussion. Therefore, we manually broke the symmetry in the revised manuscript, setting all magnetization of Co atoms as up and that of Fe atoms as down, according to published works (*J. Phys.: Condens Matter.* 2011, 23(42): 426004). For termination of CoFe₂O₄ (110), we chose the present surface where ions in octahedral vacancies were exposed based on experience of Co₃O₄ modelling (*ACS Catal.*, 2016, 6(8): 5508-5519). We have appended all these information in corresponding computational details in supplementary information.

Figure R12. (a) Temperature-Magnetization graph of CoFe_2O_4 support. Reproduced from *J. Magn. Magn. Mater.* 2009, 321(9): 1251-1255. (b) spin-alignment of CoFe_2O_4 , Fe and its spin moments are marked in purple, Co in blue.

As for the inconsistency of Pt-O coordination number between DFT models and EXAFS measurement, it is because of the special property of spinel-kind support, that is, there is always a significant portion of deposited atoms anchored on subsurface (*Nat. Commun.*, 2020, 11, 1263) confirmed by the Ar^+ sputtering experiment in our case (**Table R7**). Moreover, XAS is a characterization technique giving overall information of the catalyst entity, thus the coordination number of Pt-O has been averaged by counting all the Pt-O pairs in the sample. Therefore, it is less accurate to judge our surface model. Instead, HAADF-STEM is straightforward for obtaining information of surface. As we have mentioned above, on CoFe_2O_4 (110) surface, it is nearly impossible for Pt reaching O CN more than 4.

Table R7. The concentration of Pt on the Pt_1/CF and $\text{Pt}_1/\text{CF-W}$ obtained from XPS.

Sample	The concentration of Pt (%)	
	Pt_1/CF	$\text{Pt}_1/\text{CF-W}$
Raw	1.04	1.12
Sputtered-5 min	0.47	0.48
Sputtered-90 min	0.31	0.30

“11. The relative energetics of the molecular adsorption of water (-11103.3552 a.u.) and dissociative adsorption of water (-11103.2587 a.u.) on the CoFe_2O_4 (110) surface suggest molecular adsorption is much more favorable. Any discussion based the high

energy structure is irrelevant since water will be predominantly adsorbed in the molecular form.”

Response:

Thanks for this question. After setting support as ferrimagnetic phase, we re-performed our calculation and the results were shown in **Figure R13-15** and **Table R8**. Here the suffix “d” indicates water dissociative adsorption mode and “m” indicates water molecule adsorption mode. Obviously, the dissociative adsorption of water on the Pt/CF(110) is more energetically favorable.

Figure R13. Structure of Pt/CF model.

Figure R14. Structures of Pt/CF-W-d models. “d” indicates water dissociative adsorption.

Figure R15. Structures of Pt/CF-W-d models. “m” indicates water molecule adsorption.

Table R8. Energies and Bader charges of models.

Model	Energy/a.u.	Bader charge I*/e	Bader charge II*/e
Pt/CF	-	0.87	0.91
Pt/CF-W-d1	-11108.44	0.65	0.73
Pt/CF-W-d2	-11107.77	0.63	0.66
Pt/CF-W-d3	-11108.63	0.84	0.89
Pt/CF-W-d4	-11107.86	0.43	0.48
Pt/CF-W-d5	-11108.37	0.66	0.71
Pt/CF-W-m1	-11107.82	0.61	0.71
Pt/CF-W-m2	-11107.70	0.57	0.64
Pt/CF-W-m3	-11107.71	0.52	0.59
Pt/CF-W-m4	-11107.99	0.54	0.61
Pt/CF-W-m5	-11107.91	0.56	0.62

* “Bader charge I” is calculated by (1) partition electron density to get number of electrons populated on Pt atom (2) Subtract number of electrons from 18 (the number of electrons treated as valence electrons); “Bader charge II” is calculated by directly partitioning total charge density.

“12. The work function decreases from 3.8 eV in Pt₁/CF to 3.5 eV in of Pt₁/CF-W after water treatment as shown in Figure S19. However, the similar decrease of work function (~ 0.3 eV) is observed in water treatment of CoFe₂O₄. Therefore, the decrease of water function could be just due to the interaction between water and CoFe₂O₄ surface with nothing to do with the protonation of Pt-O bond. In fact, the calculation with a protonated oxo-bridge results in Pt-O bond distance increase from 1.91 to 1.99 Å, which is quite significantly. However, the bond distance of Pt-O extracted from EXAFS simulation in Table S2 is almost identical for Pt₁/CF and Pt₁/CF-W, 1.99 and 2.00 Å, respectively. The inconsistency between the Pt-O bond lengths question the relevance of the protonated Pt-O-Fe model, therefore undermine the hypothesis of H⁺-decorated model of Pt₁/CF-W after water treatment. In addition, as shown in Table S3, the change of work function shows strong dependence of the models. One can get work function change in a wide range with different Pt/CF(110)-H₂O models.”

Response:

Thanks for your question. We fully agree with the reviewers that the water is interacted with the CoFe₂O₄ from the work function data. This is also our point of view in this work that it is the change of CoFe₂O₄ that finally influenced the metal-support interaction and electron transfer between Pt and CoFe₂O₄. The importance of CoFe₂O₄ can also be illustrated by the fact that water can be dissociated while on other oxides such as MgAl₂O₄, the water treatment effect is negligible.

However, to release reviewer’s concern and avoid any possible misunderstanding, in the revised manuscript we don’t use work function to examine the rationality of our models anymore and only energy issue was considered in the modeling. Based on our newly obtained structures, after protonation a slight lengthening of Pt-O bond can still be observed, from (2.02, 1.97, 1.89, average 1.96) to (2.02, 2.01, 2.05, 2.07, average 2.03) Angstrom. A slight lengthening of Pt-O bond indicated a bond order decrease i.e., a weakening in metal-support interaction (see **Table R6** in answer to the 5th

question). As we mentioned above, because EXAFS is not sensitive to surface structure change, the bond length measured is an averaged value between surface Pt-O and subsurface Pt-O, it will not be accurate enough to make direct and reasonable comparison with simulated results.

“13. What is Pt(Ac)/CF-W in Figure S20? Should it be Pt(Ac)/CF?”

Response:

We are sorry for the wrong label in Figure S20. It is Pt(Ac)/CF. We have corrected the mistake and thoroughly examined the supporting information.

“14. No error bars of the experimental data have been provided.”

Response:

We sincerely thank the reviewer for raising this issue. As suggested by the reviewers, we have added the error bars of all the activity evaluation experiment (**Figure R10-11**). We would like to declare that the reproducibility of both catalyst synthesis and water treatment effect is very good. We have never encountered the reproducibility issue in the past 4 years for many times experiments.

*“15. Spell out the acronyms in figure caption, especially when they have not been explanation in the text proceeding the corresponsable figures, e.g. GHSV in Figures 3 and 5 has not been defined until line 333 on page 14. ‘WT’ in the caption of Figure S5. Why are the GHSVs differ so much in different experiments? GHSV = 40, 000 ml/gcat*h in Figure 3 and 5, GHSV = 1000 L/g Pt *h in Figure S11, GHSV = 20,000 ml/gcat*h in Figure S12, GHSV = 20,000 ml/gcat*h in Figure S8, GHSV = 426,000 ml/mmolmetal*h in Figure S7. Any specific reasons to choose those GHSV?”*

Response:

We are sorry for presenting GHSV in different units. Actually, the velocity of feed gas (0.5 vol% CH₄, 20 vol% O₂, 79.5 vol% N₂) was 33 mL/min for all the light-off test excepted specific declaration. Generally, 100 mg 1 wt% Pt loading of the catalysts or 50 mg 2 wt% Pt loading of the catalysts were tested in the light-off experiment. For the kinetic experiments, the GHSV was varied to keep the conversion

below 15 % in the give temperature range. We have unified the GHSV and sorry for any inconvenience.

We genuinely thank this reviewer again for his/her patient and dedicated to our manuscript. As suggested, we have double-checked the manuscript and improved the quality of English writing in the revised manuscript.

Reviewer #3

“The paper reports on the CMSI of Pt/CoFe₂O₄ weakened by simple water soaking treatment has a positive effect on the C-H bond activation during the methane combustion reaction. The authors devised a method to maximize the effect of a single atom with a simple treatment method, and this was proven through various characterizations. However, the direct reason why small structural differences, such as the decoration of H⁺ at the Pt-O-Fe interface, actually boosting the activity significantly, is not clearly explained analytically. The authors should report the apparent activation energies and longevity tests for the catalysts to show whether this difference activity is a transient phenomenon.”

Response:

We sincerely thank the reviewer for raising this issue. For the question of why small structural differences actually boosting the activity significantly, we found that after water treatment, the activation of methane over less oxidized Pt was greatly enhanced from the results of CH₄ pulse reaction and CH₄-TPR experiment (**Figure 5a and Table R4**). Since the activation of first C-H bond in methane is the rate-limiting step in the methane combustion, the increase capability of Pt₁ in CH₄ activation directly reflected on its activity promotion. To further understand the structural change of Pt single atoms, we performed CO-DRIFTS, H₂-TPR and DFT calculation, of which the results suggest change of vicinal coordination environment of Pt single atoms (SAs) by the dissociating of H₂O molecular on the CoFe₂O₄ thus lowering the chemical state of Pt SAs with increasing redox ability.

As suggested by the reviewers, we have conducted kinetic experiments to measure the apparent activation energies. As displayed in **Figure R16**, the activation energy is 92 kJ/mol for Pt₁/CF catalysts. As expected, the water treatment results in a decrease in the activation energy of Pt₁/CF catalysts, with a value of 77 kJ/mol.

Figure R16. Arrhenius-type plots of methane combustion kinetics. All experiments were conducted with 0.5 vol% CH₄, 20 vol% O₂, and 79.5 vol% N₂. The GHSV was 20,000 for Pt₁/CF; the GHSV was 80,000 for Pt₁/CF-W.

We provided the stability test of Pt₁/CF-W under the wet conditions (2.3 vol% H₂O) in the previous manuscript (**Figure S13**). Despite the fact that the addition of water suppressed the conversion of methane, the Pt₁/CF-W showed excellent stability in the presence of steam without deactivation. By contrast, the catalyst slowly deactivated under dry feed gas most likely due to the dehydration process and the enhanced CMSI meanwhile. More extensive stability testing of the Pt₁/CF-W in dry feed gas has been conducted which showed a maintained activity after a rapid decrease in the first 5 h (**Figure R17**). Note that the conversion of methane over Pt₁/CF under the same conditions is below 1 wt%, Pt₁/CF-W still owns great superiority over Pt₁/CF. This further verifies that the enhanced activity is not a transient phenomenon.

Figure R17. Methane conversion as a function of time on Pt₁/CF-W catalyst.

The reaction conditions: 0.5 vol% CH₄, 20 vol% O₂, and 79.5 vol% N₂, GHSV = 20,000 ml/(g_{cat}·h), 300 °C.

We have added the apparent activation energies data and the long-time stability test in the supporting information (Figure S9 and S14) and relevant discussion in the revised manuscript on page 7 and 8, respectively.

REVIEWERS' COMMENTS

Reviewer #1 (Remarks to the Author):

I have confirmed that the paper was successfully revised according to the referees' comments. For the comment 1, the authors successfully demonstrated unchanged Pt sites by means of ion scattering spectroscopy (Figure S4). For the comment 2, the authors examined solubility of the catalyst in water and also added XPS results in Table S5. For the comment 3, the authors added the Table R6. For the comments from the reviewers 2 and 3, they reasonably answered and revised the manuscript. I recommend the publication of this paper.

Reviewer #2 (Remarks to the Author):

The authors have performed additional experiments to further confirm that their Pt/CoFe₂O₄ systems is indeed singlet atom catalysts, and the excellent performance improvement is due simply due to the reaction of water with the Pt/CoFe₂O₄ SAC. The simple treatment, the beautiful performance, and its potential applications to other SACs make it suitable to publish in Nature Communications. However, the evidence to support the coordination H⁺ is still not convincing. It may be better to state this be one of the possible mechanism instead a conclusive one as claimed in the current version of paper. In addition, the comparison between dissociative and molecular adsorption of water is not a fair comparison. Many of the dissociative water adsorption models have three-coordinated Pt. But in the molecular adsorption model, all models has two-coordinated Pt, which could result in high-energy structure due to the highly under-coordinated Pt.

The simplicity, beautiful experiments and excellent performance ensure the publication of this paper. However, interpretation of the experimental results should allow other possibility besides H⁺-coordination and water dissociation.

Reviewer #3 (Remarks to the Author):

The authors are well revised my comments. Since the author used commercial CoFe₂O₄, a minor comment I raised is that impurities such as alkalis and metal ions are likely to present. These impurities may be migrated or shuffled to Pt during the water soaking step and may affect the reactivity of Pt minimizing structural influence. If possible, it would be good to report whether surface impurities remain in CoFe₂O₄. In Figure R11, the CoFe₂O₄-W is more reactive than CoFe₂O₄, so it is good to verify whether the effect of impurities on the support is not critical. Indirectly, this can be demonstrated by preparing a Pt catalyst using CoFe₂O₄ pre-washed clearly with D.W. water(not soaking) as a support and testing this catalyst.

Point-by-point response to reviewers' comments

Manuscript ID: NCOMMS-21-43244

Title: Modulating the strong metal-support interaction of single-atom catalysts via vicinal structure decoration

Reviewer #1

“I have confirmed that the paper was successfully revised according to the referees' comments. For the comment 1, the authors successfully demonstrated unchanged Pt sites by means of ion scattering spectroscopy (Figure S4). For the comment 2, the authors examined solubility of the catalyst in water and also added XPS results in Table S5. For the comment 3, the authors added the Table R6. For the comments from the reviewers 2 and 3, they reasonably answered and revised the manuscript. I recommend the publication of this paper.”

We sincerely thank the reviewer for his/her approval.

Reviewer #2

“The authors have performed additional experiments to further confirm that their Pt/CoFe₂O₄ systems is indeed singlet atom catalysts, and the excellent performance improvement is due simply due to the reaction of water with the Pt/CoFe₂O₄ SAC. The simple treatment, the beautiful performance, and its potential applications to other SACs make it suitable to publish in Nature Communications. However, the evidence to support the coordination H⁺ is still not convincing. It may be better to state this be one of the possible mechanism instead a conclusive one as claimed in the current version of paper. In addition, the comparison between dissociative and molecular adsorption of water is not a fair comparison. Many of the dissociative water adsorption models have three-coordinated Pt. But in the molecular adsorption model, all models has two-coordinated Pt, which could result in high-energy structure due to the highly under-coordinated Pt. The simplicity, beautiful experiments and excellent performance ensure the publication of this paper. However, interpretation of the experimental results should allow other possibility besides H⁺-coordination and water dissociation.”

We sincerely thank the reviewer for his/her suggestion. We agree that the decoration of H^+ is one of the hypotheses considered based on the evidence we have at current stage. In this work, we aim at reporting the water treatment effect on the activity and metal-support interaction of SACs. The mechanism that how the adsorption of water changes the metal-support interaction leaves an open question to be further studied.

Reviewer #3

“The authors are well revised my comments. Since the author used commercial CoFe2O4, a minor comment I raised is that impurities such as alkalis and metal ions are likely to present. These impurities may be migrated or shuffled to Pt during the water soaking step and may affect the reactivity of Pt minimizing structural influence. If possible, it would be good to report whether surface impurities remain in CoFe2O4. In Figure R11, the CoFe2O4-W is more reactive than CoFe2O4, so it is good to verify whether the effect of impurities on the support is not critical. Indirectly, this can be demonstrated by preparing a Pt catalyst using CoFe2O4 pre-washed clearly with D.W. water(not soaking) as a support and testing this catalyst.”

We sincerely thank the reviewer for his/her valuable comments. As suggested by the reviewer, we have washed the commercial support (2 g) with a large amount of water (3 L) to reduce the potential impurities before uploading Pt metal. After that the oxide was calcined at 500 °C for 4 h followed by the impregnation of H_2PtCl_6 . The preparation and treatment procedure were exactly the same as we did before. The catalysts prepared using the purified support were labeled as Pt₁/CF_P and Pt₁/CF_P-W, respectively. Their activity results are shown in the **Figure R1**. Clearly a same scenario was observed that the activity increased significantly after water treatment, excluding the possible effect of purity of the support and further verifying the importance of water treatment after loading of metal, i.e., the effect on metal-support interaction.

Figure R1. The light-off curves of CH₄ combustion. Reaction conditions: 0.5 vol% CH₄, 20 vol% O₂, and 79.5 vol% N₂, GHSV = 20,000 ml/(g_{cat}·h).

We have added this new data in the supporting information (Figure S18b) and the corresponding discussion into the revised manuscript on page 9.